Inhibition of Connexin 43 reverses ox-LDL-mediated inhibition of autophagy in VSMC by inhibiting the PI3K/Akt/mTOR signaling pathway

Qin Xuqing 1 2 3
He Wenjun 2 3 4
Yang Rui 1 2 3
Liu Luqian 2 3 4
Zhang Yingying 1 2 3
Li Li 2
Si Junqiang 1 2 3
Li Xinzhi 2 3 4 lixinzhi@shzu.edu.cn
Ma Ketao 1 2 3 maketao@hotmail.com
1 Shihezi University School of Medicine, Department of Physiology , Shihezi, Xinjiang , China
2 Ministry of Education, Shihezi University School of Medicine, Key Laboratory of Xinjiang Endemic and Ethnic Diseases , Shihezi, Xinjiang , China
3 First Affiliated Hospital, Shihezi University School of Medicine, NHC Key Laboratory of Prevention and Treatment of Central Asia High Incidence Diseases , Shihezi, Xinjiang , China
4 Shihezi University School of Medicine, Department of Pathophysiology , Shihezi, Xinjiang , China
Moraczewska Joanna
Electronic publication date: 2022 Mar 16
Publication date: 2022
Volume: 10
Electronic Location ID: e12969
Received 2021 Jul 30; Accepted 2022 Jan 30
Copyright: © 2022 Qin et al.
Copyright year: 2022
Copyright holder: Qin et al.
License: This is an open access article distributed under the terms of the Creative Commons Attribution License, which permits unrestricted use, distribution, reproduction and adaptation in any medium and for any purpose provided that it is properly attributed. For attribution, the original author(s), title, publication source (PeerJ) and either DOI or URL of the article must be cited.
License URL: https://creativecommons.org/licenses/by/4.0/

Keywords: Atherosclerosis, Vascular smooth muscle cells, Autophagy

Funding: National Natural Science Foundation of China 81860085, 81860286 Corps Science and Technology Cooperation Project of China 2020BC004 Central Research Institute Fund of Chinese Academy of Medical Sciences 2020-PT330-003 This work was supported by the National Natural Science Foundation of China (Nos. 81860085, 81860286), Corps Science and Technology Cooperation Project of China (No. 2020BC004), Central Research Institute Fund of Chinese Academy of Medical Sciences (2020-PT330-003). The funders had no role in study design, data collection and analysis, decision to publish, or preparation of the manuscript.

==============================
Background

Oxidized low-density lipoproteins (ox-LDL) may induce foam cell formation from the vascular smooth muscle cell (VSMC) by inhibiting VSMC autophagy. This process accelerates the formation of atherosclerosis (AS). Connexin 43 (Cx43), which is the most widely distributed connexin in VSMC is associated with autophagy. However, the mechanism of action and the involvement of Cx43 in ox-LDL-inhibited VSMC autophagy remain unclear.

Methods

The primary VSMC were obtained and identified, before primary VSMC were pretreated with an inhibitor (Cx43-specific inhibitor Gap26 and PI3K inhibitor LY294002) and stimulated with ox-LDL.

Results

Ox-LDL not only inhibited autophagy in VSMC via downregulation of autophagy-related proteins (such as Beclin 1, LC3B, p62), but also increased Cx43 protein levels. Then we added Gap26 to VSMC in the ox-LDL+Gap26 group, in which autophagy-related proteins were increased and the accumulation of lipid droplets was reduced. These result suggested that an enhanced level of autophagy and an alleviation of lipid accumulation might be caused by inhibiting Cx43 in VSMC. The phosphorylation levels of PI3K, AKT, mTOR were increased by ox-LDL, thus down-regulating autophagy-related proteins. However, this situation was partially reversed by the Gap26. Moreover, Cx43 expression were decreased by LY294002 in ox-LDL-induced VSMCs.

Conclusion

Inhibiting Cx43 may activate VSMC autophagy to inhibit foam cell formation by inhibiting the PI3K/AKT/mTOR signaling pathway.

Introduction

Atherosclerotic cardiovascular disease is the main contributor to the increase in global morbidity and mortality (Arnett et al., 2019). Atherosclerotic plaques are formed in the vessel wall due to the accumulation of lipids and other substances in the process of atherosclerosis (AS) (Poznyak et al., 2020). These lipid-rich cells in atherosclerotic plaques are called foam cells (FCs). Many past studies suggested that the majority of FCs originated from macrophages. However, recently it has been shown that most of the FCs in mouse and human atherosclerotic plaques originate from smooth muscle cell (SMC) (Wang et al., 2019a; Allahverdian et al., 2014). Accordingly, understanding the mechanism of SMC-derived FCs will contribute to the prevention and treatment of atherosclerotic diseases.

Autophagy is a physiological process in which eukaryotic cells maintain internal homeostasis (Qin, 2019). It is also involved in various physiological and pathological processes, including hypertension and the formation of coronary atherosclerotic plaques (Hughes, Beyer & Gutterman, 2020). In human atherosclerotic plaques, autophagy may be induced in VSMC, macrophages, and endothelial cells stimulated by lipids, metabolic stress, cytokines and reactive oxygen species (Grootaert et al., 2018). Over time, almost all types of cells have a certain degree of autophagy dysfunction, which negatively impacts plaque progression, but the underlying mechanisms are different (Grootaert et al., 2018). In addition, more studies have proven that regulation of autophagy may improve the occurrence and development of AS (Kumar et al., 2020; Shi et al., 2020). Rosuvastatin has been shown to enhance autophagic activity in macrophages by inhibiting the activation of the PI3K/Akt/mTOR signaling pathway and increasing autophagic flux, thus achieving the anti-AS effect (Zhang et al., 2021). The PI3K/Akt/mTOR signaling pathway is a major regulatory pathway that inhibits autophagy initiation. The PI3K/AKT/mTOR signaling pathway is a classical autophagic pathway. Meanwhile, it was also found that Danlou tablets (DLTs) can promote autophagy in macrophage by inhibiting this signaling pathway, thereby reducing FCs formation and improving AS (Liu et al., 2021). Despite lots of studies on the role of autophagy in anti-AS, those on the specific mechanism of autophagy is limited.

Gap junction channel (GJC), which is composed of connexin (Cx) as the basic unit, is a special channel to mediate direct communication at the intercellular (Beyer & Berthoud, 2018). Cx43 is the major Cx forming GJ in SMCs (Morel & Kwak, 2012). Gap26 is a specific Cx43 mimic peptide (Qing et al., 2021) that may inhibit Cx43 expression (Li et al., 2015, 2017) when used as a Cx43-GJ inhibitor. The increased expression of connexin 43 in intimal VSMC was observed in the early stages of human and mouse atherosclerotic lesions (Leybaert et al., 2017). This effect may be associated with various risk factors for AS, such as ox-LDL (Wang et al., 2019b) or angiotensin II (Lei et al., 2019), which may upregulate the expression of Cx43. Recently, studies have demonstrated that there is an important interaction between Cx and autophagy, that is Cx could be used not only as an autophagy substrate, but also an autophagy regulator (Iyyathurai et al., 2016). Besides that, it is reported that the regulation of Cx43 has impacts on the phosphorylation level of proteins in some signaling pathways, such as PI3K/Akt signaling pathway and related downstream pathways (Ji et al., 2019; Wang et al., 2020). This finding suggests that Cx43 and PI3K/Akt-related downstream pathways may be interrelated. Nevertheless, the connection between Cx43 in VSMC and autophagy remains elusive in atherosclerotic lesions. Therefore, it is worth exploring whether Cx43 takes part in regulating the PI3K/AKT/mTOR signaling pathway to have an impact on autophagy and understanding the role of Cx43 in autophagy may reveal its importance as a regulatory mechanism of disease development, making it a potential therapeutic target.

Materials and Methods

Culture of thoracic aortic VSMC and study design

Male Sprague-Dawley (SD) rats, 6–8 weeks of age (210–250 g), were selected from the Animal Experimental Center of Xinjiang Medical University. The license number was SCXK Xin (2018-001). All animals were treated humanely. Experimental research strictly conformed to the regulations of the Medical Ethics Committee of Shihezi University.

High-purity VSMC were then obtained from the thoracic aortas of the rats (drug anaesthesia, 2% pentobarbital sodium (40 mg/kg), intraperitoneal injection) by in vitro culture, as described previously (Owens et al., 1986). The rats were euthanized via intravenous overdose of pentobarbital sodium. Third- to sixth-passage (P3–P6) VSMC were used in all experiments and cultured with DMEM/F12 (10% FBS).

Ox-LDL (80 μg/ml) was treated to primary VSMC for periods of 0, 3, 6, 12, 24 and 48 h and different concentrations of ox-LDL (0, 20, 40, 80, 160 μg/ml) were incubated for 24 h. The experiment was then divided into four groups, including the Control group, ox-LDL (Li et al., 2014) (80 μg/ml, 24 h) group, ox-LDL + Gap 26 (Wang et al., 2019b) (Cx43 specific inhibitor, 100 μM pretreatment for 45 min, and then ox-LDL for 24 h,) group, ox-LDL + LY294002 (Kim et al., 2020) (20 μM pretreatment for 30 min and then ox-LDL for 24 h) group, and the Gap 26 group.

Reagents and antibodies

ox-LDL (Cat. No. YB-002, Yiyuan Biotechnology, Guangzhou, China), TRIzol reagent (Thermo Fisher Scientific, Inc., Waltham, MA, USA), Gap26 (Cat. No. A1044; APExBIO Technology LLC, Houston, TX, USA), LY294002 (Cat. No. A8250, APExBIO Technology LLC, Houston, TX, USA), anti-α-SMA (ab124964), anti-Desmin (ab32362), anti-LC3B (ab192890), anti-p62 (ab109012), the secondary antibody of western blotting were obtained from ZSGB-BIO (Beijing, China), the secondary antibody of immunofluorescence were obtained from ZSGB-BIO (Beijing, China), polyclonal rabbit anti-Beclin 1 (ab62557), and anti-Cx43 (ab11370) were obtained from Abcam (Cambridge, UK). Anti-PI3K p85 (#4257), anti-AKT (#4060), anti-mTOR (#2983), anti-phospho-PI3K p85 (Tyr458)/p55 (Tyr199)-actin (#17366), anti-phospho-AKT (#13038), and anti-phospho-mTOR (#5536) were obtained from CST (Inc).

Western blot analysis

The proteins from each VSMC samples were lysed after their respective drug treatments. The protein concentration was measured with BCA assays. The proteins were separated on 10% or 12% SDS-PAGE, electroblotted onto PVDF membranes, and blocked with 5% BSA, followed by immunoblotting. The primary (1:1,000) and secondary (1:10,000) antibodies were incubated in order. Finally, image analysis was performed using ImageJ software (National Institutes of Health, Bethesda, MD, USA).

Immunofluorescence analysis

After the VSMC samples were fixed with 4% paraformaldehyde for 15 min, permeabilized with 0.2% Triton X-100 for 3 min and blocked by 5% BSA for 30 min, they were then incubated with primary (1:100) and secondary (1:100) antibodies. Subsequently, DAPI (1:1,000, Sigma-Aldrich, MerckKGaA, Burlington, MA, USA) was counterstained for 10 min to display where the nucleus is. Images were acquired using laser confocal microscopy (Zeiss LSM 510 META, Jena, Germany).

Quantitative real-time PCR

The total RNA from VSMC sample was isolated using TRIzol reagent. Next, cDNA was synthesized using a kit (Thermo Fisher Scientific, Inc., Waltham, MA, USA). The gene primers used were:

Cx43Forward:TCACGTCCCACGGAGAAAACReverse:ATCCGCAGTCTTTTGATGGGGAPDHForward:GACATGCCGCCTGGAGAAACReverse:AGCCCAGGATGCCCTTTAGT

SYBR Green Real-time PCR master mix (Toyota Corporation, Toyota City, Japan) was used to amplify the cDNA in the reaction system, which consisted of 40 cycles of 50, 95, and 60 °C. The 2–ΔΔCt method was used to analyze the data (Livak & Schmittgen, 2001).

Transmission electron microscopy (TEM)

After the cells terminated digestion, the bottom cells were left to add 2.5% glutaraldehyde (special for electron microscopy) at 4 °C overnight. After that, autophagosomes were observed by transmission electron microscopy (JEOL, Tokyo, Japan).

Oil red O staining

The accumulation of lipid droplets was determined by referring to the instructions of the oil red staining kit (Cat. No. D027-1, NJJCBIO, Nanjing, China).

Statistical analysis

GraphPad 6.01 software was used for statistical analysis and chart preparation and the results are expressed as mean ± SD. One-way analysis of variance (ANOVA) was used to compare the differences among groups, and the Student’s test was used for pairwise comparisons between groups. A value of P < 0.05 was considered to be statistically significant.

Results

Ox-LDL inhibited autophagy in VSMC via downregulation of autophagy-related protein

To examine the effect of ox-LDL on autophagy in VSMC, we monitored changes in the autophagy marker proteins p62, Beclin1, and LC3B, after they were treated with various concentrations of ox-LDL, and ox-LDL was used at different lengths of time. In our study, ox-LDL decreased Beclin-1 protein levels and the ratio of LC3II to LC3I at various time points in VSMC while the p62 protein levels were increased (Figs. 1A–1D). The p62 levels were increased with the elevation of ox-LDL concentrations, which was contrary to the results of Beclin 1 and LC3B (Figs. 1E–1H). These data indicated that ox-LDL inhibited autophagy in VSMC in a time-and concentration-dependent manner.

Figure 1 The effect of various concentrations of ox-LDL for different lengths of time on autophagy marker proteins in VSMCs.

(A–H) Immunoblot analysis of p62, Beclin 1 and LC3B, after ox-LDL (80 μg/ml) was treated at various time (A–D) points (0, 3, 6, 12, 24 and 48 h) and different doses (E–H) of ox-LDL (0, 20, 40, 80, 160 μg/ml) were incubated for 24 h. (*P < 0.05 vs. Control, **P < 0.01 vs. Control, n = 3, data shown as the mean ± SD).

Ox-LDL increased Cx43 protein levels in VSMC

Next, we investigated the effect of ox-LDL on Cx43 in VSMC. The results appeared that ox-LDL treatment (80 μg/ml, 24 h) obviously elevated the protein (Figs. 2A and 2B) and mRNA (Fig. 2C) levels of Cx43 in VSMC. Moreover, the semi-quantitative results of immunofluorescence appeared that the relative fluorescence intensity of Cx43 were evidently enhanced after ox-LDL treatment, and Cx43 was mainly distributed in the cytoplasm and cell membrane (Figs. 2D and 2E). Thus, the above results prompted us that ox-LDL-inhibited autophagy in VSMC was accompanied by the upregulation of Cx43 protein levels.

Figure 2 The effect of ox-LDL on the expression and localization of Cx43 in VSMCs.

(A and B) Immunoblot analysis of Cx43 on ox-LDL-treated (80 μg/ml, 24 h) VSMC. (C) qRT-PCR analysis for Cx43 mRNA levels in VSMC. (D and E) Immunofluorescence analysis of Cx43 (green), DAPI (blue) staining of the nucleus. Scale bar = 50 μm. (*P < 0.05 vs. Control, **P < 0.01 vs. Control, n = 3, data shown as the mean ± SD).

Involvement of Cx43 in VSMC autophagy and FCs formation mediated by Ox-LDL

To identify whether Cx43 was involved in ox-LDL-inhibited autophagy in VSMC, we assessed the role of Gap26, an Cx43-specific inhibitor, on autophagy. The results demonstrated that Gap26 treatment increased Beclin 1 protein levels and the induction of LC3II in VSMC incubated with ox-LDL, which was the opposite of p62 protein levels (Figs. 3A–3D). And the immunofluorescence results were consistent with those of the Western blot (Figs. 3E–3J). At the ultrastructural level, autophagosomes presents a double-membraned structure, which contains undigested cytoplasmic contents. The TEM studies indicated that Gap26 treatment increased the number of autophagosomes compared with that in the ox-LDL group (Fig. 3K). These results revealed that inhibiting Cx43 could improve the autophagy level of VSMC. To clarify whether alternation of autophagy levels affected transformation of VSMC into foam cells, we evaluated the changes in lipid droplets (Fig. 3L). The findings suggested that inhibition of Cx43 prevented the ox-LDL-induced lipid droplet accumulation in VSMC, thereby suppressing FCs formation.

Figure 3 The function of Cx43 in ox-LDL-inhibited autophagy in VSMCs.

(A–D) Immunoblot analysis of p62 (B), Beclin 1 (C) and LC3B (D) in ox-LDL-mediated VSMCs after Gap26 pretreatment. (E–J) Immunofluorescence analysis of p62 (E and H), Beclin 1 (F and I) and LC3B (G and J) in VSMCs. DAPI (blue) staining of the nucleus. Scale bar = 50 μm. (K) The changes in autophagosome were detected by TEM. The yellow arrow indicated autophagosomes. Scale bar = 2 μm. (L) The accumulation of lipid droplets in VSMCs was examined by oil red O staining. Scale bar = 100 μm. (*P < 0.05 vs. Control, **P < 0.01 vs. Control, #P < 0.05 vs. ox-LDL, ##P < 0.01 vs. ox-LDL, n = 3, data shown as the mean ± SD).

Inhibition of Cx43 up-regulated the autophagy levels of VSMC after ox-LDL treatment via inhibiting the PI3K/AKT/mTOR signaling pathway

We pretreated VSMC with LY294002 and Gap26 to explore the potential mechanism by which Cx43 is involved in ox-LDL-inhibited autophagy in VSMC. As compared with ox-LDL-treated VSMC, LY294002 markedly inhibited p62 protein levels, increased Beclin-1 protein levels, and promoted the conversion of LC3-I to LC3-II (Figs. 4A–4D), indicating that PI3K/AKT/mTOR signaling pathway might be involved in ox-LDL-mediated autophagy of VSMC. Next, ox-LDL up-regulated the protein levels of p-PI3K, p-AKT, and p-mTOR, which was partially reversed by Gap26 (Figs. 4E–4H), suggesting that Cx43 inhibition prevented the activation of this pathway. To further analyse the relationship between Cx43 and this signaling pathway, LY294002 treatment was used to examine changes in Cx43 protein levels in ox-LDL-treated VSMC. Cx43 protein levels was decreased as VSMC was exposed to LY294002 (Figs. 4I and 4J).

Figure 4 The relationship between Cx43 and PI3K/Akt/mTOR.

(A–D) Immunoblot analysis of p62 (B), Beclin 1 (C) and LC3B (D) in VSMCs treated with ox-LDL after LY294002 pretreatment. (E–H) Immunoblot analysis of the marker proteins of PI3K/AKT/mTOR signaling pathway in VSMCs after Gap pretreatment. (I and J) Immunoblot analysis of Cx43 in ox-LDL-treated VSMCs after LY294002 pretreatment. (*P < 0.05 vs. Control, **P < 0.01 vs. Control, #P < 0.05 vs. ox-LDL, n = 3, data shown as the mean ± SD).

Discussion

VSMC are an important source of FCs in atherosclerotic lesions and have received increased attention in recent years. Many studies have reported that FCs formation may be inhibited by regulating autophagy in atherosclerotic lesions (Kumar et al., 2020; Shi et al., 2020; Hui et al., 2021). We demonstrated that ox-LDL-inhibited autophagy induction in VSMC via the downregulation of autophagy-related protein. Cx43 protein levels were also found to increase after ox-LDL treatment. The inhibition of Cx43 could activate autophagy and reduced ox-LDL-mediated lipid droplet accumulation by inhibiting PI3K/Akt/mTOR signaling pathway.

There are many risk factors for the formation of AS, and one of which is ox-LDL (Khatana et al., 2020). It is closely associated with the formation of FCs. Consequently, we used the atherogenic factor ox-LDL (Zhao et al., 2020) to induce the formation of VSMC-derived FCs. There is increasing in vitro evidence that autophagy is present in atherosclerotic plaques (Martinet & De Meyer, 2009). However, autophagy has dual effects on AS (Martinet & De Meyer, 2008, 2009) and may be either protective or harmful. Therefore, we examined the effect of ox-LDL on VSMC autophagy and found that it regulated autophagy in a time-dependent manner (Li et al., 2014). Additionally, the effects of different doses of ox-LDL on autophagy in SMC were also different (Ding et al., 2013). In the early stage of AS, a low concentration of ox-LDL increased the protective autophagy in SMCs, while a high concentration of ox-LDL weakened the protective effect of autophagy and enhanced autophagy-induced cell death (Hassanpour et al., 2019). ox-LDL downregulated the accumulation of LC3-II and Beclin1 protein levels in macrophage transformation into the foam cell (Cao et al., 2019). Nevertheless, ox-LDL was found to increase the LC3II/LC3I ratio and Beclin1 protein levels in vascular endothelial cells in a cell injury model of vascular endothelial cells induced by ox-LDL (Li et al., 2018). This finding implied that there were differences in the influence of ox-LDL on autophagy in different cell models.

We also determined that ox-LDL increased Cx43 expression on VSMC. However, few studies have shown a relationship between Cx43 and autophagy in VSMC and the roles of these factors. It has been demonstrated that Cxs could regulate autophagy (Iyyathurai et al., 2016). Cx43 located on the plasma membrane may directly interact with autophagy-related gene 16 (Atg16), which is the initial step in the formation of autophagosomes. Therefore, Cxs may act as negative regulators of basal autophagy by binding to the components of the initiation complex. Other Cx subtypes (Cx26 or Cx32) may also have negative effects on autophagic flux, suggesting that some members of the Cx family may regulate autophagy. Pharmaceuticals including tamoxifen and lindane have been shown to increase autophagy in Cx43-expressing cells but not in Cx43-knockout cells (Iyyathurai et al., 2016), which suggests that Cx43 may take part in the regulation of autophagy. Other studies have found the mRNA level of LC3B in Cx43-knockout group was significantly increased compared with untreated porcine early embryos (Shin et al., 2020). However, there have been few reports on the interaction between Cx43 and autophagy in AS (Iyyathurai et al., 2016). Our results showed that the autophagic flux in VSMC was impaired after ox-LDL incubation, and this effect was partly reversed by the specific Cx43 inhibitor, Gap26.

Furthermore, studies have demonstrated that autophagy takes part in lipid metabolism. Autophagy may be effective in the decomposition of lipid droplets to provide free fatty acids for cells (Martinez-Lopez & Singh, 2015). The role of autophagy in lipid metabolism is associated with triglycerides breakdown and cholesterol stored in lipid droplets by selective lysosome-dependent macrophages (Liu & Czaja, 2013). Many important metabolic disorders, such as fatty liver, obesity, and AS, may be associated with impaired lipid autophagy (Robichaud et al., 2021; Khawar, Gao & Li, 2019). Our results show that ox-LDL-induced FC formation may be inhibited by activating autophagy (Ko et al., 2020) and that autophagy levels increased and intracellular lipid droplets decreased significantly after the inhibition of Cx43 in VSMC. This effect may be associated with the activation of autophagy and the increase of autophagosomes, which can engulf more lipid droplets and bind to lysosomes to degrade lipids.

The PI3K/Akt/mTOR signaling pathway has been demonstrated that it plays an important role in regulating autophagy in AS (Jiang et al., 2017) and our study has shown that ox-LDL could activate this pathway (Zhuo et al., 2019). Propofol reduces cell adhesion by inhibiting Cx43 and the downstream PI3K/Akt/NF-κB signaling pathway. Cx43 was overexpressed (or knocked down) on monocytes, resulting in a decrease of the related protein p-AKT with the downregulation of Cx43, and vice versa (Ji et al., 2019). Other studies have shown that resveratrol inhibits Akt and its downstream pathway by upregulating Cx43 in colorectal cancer cell lines (Wang et al., 2020). This finding suggests that there is a relationship between Cx43 and the PI3K/AKT pathway. Therefore, we used the PI3K-specific inhibitor LY294002 (Fang et al., 2020) and Gap26 to confirm whether there is a connection between Cx43 and the PI3K/AKT/mTOR signaling pathway. We found that the promoting effect of Cx43 on autophagy may be produced by inhibiting the activation of this pathway, thus preventing FCs formation. Furthermore, we found that LY294002 decreased Cx43 protein levels. This effect may be associated with the fact that when autophagy is induced (such as during hunger), internalized Cx43 will be degraded by autophagy (Catarino et al., 2020). Autophagy has been identified as an important pathway in the degradation of different types of Cxs, and autophagy induced by different factors will directly affect the expression level of Cxs (Iyyathurai et al., 2016; D’hondt et al., 2016). We hypothesized that VSMC autophagy was activated by LY294002, which resulted in the degradation of internalized Cx43, thereby reducing Cx43 expression (Bi et al., 2017). These findings suggest that Cxs can be used as autophagy substrates and as autophagy regulators.

Based on the above results, this study still has some limitations. Although some studies have reported that in podocytes (MPC5 cell line), silencing of Cx43 expression improved autophagy flow experimentally impaired by exposure of the cells to high glucose levels (Ji et al., 2019), the role of Cx43 in ox-LDL-mediated autophagy of primary VSMC still needs to be verified by more experimental techniques, such as knockdown Cx43 by transfection technology. In addition, we can also explore the role of gap junction intercellular communication (GJIC) function in VSMC autophagy. It has been found that GJIC negatively regulated autophagy and inhibited autophagy flux (Zou et al., 2020). In summary, ox-LDL prevented VSMC autophagy. The inhibition of Cx43 significantly reversed this effect. The underlying mechanism may be mediated by inhibiting the PI3K/Akt/mTOR signaling pathway. These findings may become an important regulatory mechanism for disease development, making it a potential target for the treatment of AS.

Supplemental Information

Supplemental Information 1 Culture and identification of thoracic aortic VSMCs and construction of the FC model.

(A) Growth of SMCs. (a) At 4–7 days of culture, the cells climbed out from the sides of the blocks. (b) On the 15th day of culture, the cells covered the bottom of the dish. (c) The cells exhibited typical peak-valley growth patterns; scale bar: 100 μm. (B) The expression of α-SMA and Desmin on P3 and P6 VSMCs. (C and D) Statistical analysis of α-SMA and Desmin protein expression (n = 3, P = 0.2746 and 0.7226). (E) Expression and localization of α-SMA and Desmin in P3 and P6 VSMCs. Nuclei were counterstained with DAPI (blue). Scale bar: 50 μm. (F) Identification of the FC model. After ox-LDL intervention, the number of oil red O-positive cells in primary cultured VSMCs was significantly higher than that in the control group. The black arrow indicates the red-stained lipid droplets in the cytoplasm. Scale bar: 100 μm.

Click here for additional data file.

Supplemental Information 2 Statistics of results.

Mean, SD and confidence interval.

Click here for additional data file.

Supplemental Information 3 Raw data for western blotting.

Click here for additional data file.

Supplemental Information 4 Raw numeric data.

Click here for additional data file.

Supplemental Information 5 The ARRIVE guidelines 2.0: author checklist.

Click here for additional data file.

We would like to thank the Key Laboratory of Xinjiang Endemic and Ethnic Diseases, the NHC Key Laboratory of Prevention and Treatment of Central Asia High Incidence Diseases, and the Department of Physiology and Pathophysiology of Shihezi University School of Medicine for their experimental platform and technology support.

Additional Information and Declarations

Competing Interests

Author Contributions

Animal Ethics

Data Availability

The authors declare that they have no competing interests.

Xuqing Qin conceived and designed the experiments, performed the experiments, analyzed the data, prepared figures and/or tables, authored or reviewed drafts of the paper, and approved the final draft.

Wenjun He performed the experiments, prepared figures and/or tables, and approved the final draft.

Rui Yang analyzed the data, prepared figures and/or tables, and approved the final draft.

Luqian Liu performed the experiments, analyzed the data, authored or reviewed drafts of the paper, and approved the final draft.

Yingying Zhang performed the experiments, analyzed the data, authored or reviewed drafts of the paper, and approved the final draft.

Li Li analyzed the data, authored or reviewed drafts of the paper, and approved the final draft.

Junqiang Si analyzed the data, authored or reviewed drafts of the paper, and approved the final draft.

Xinzhi Li conceived and designed the experiments, analyzed the data, prepared figures and/or tables, authored or reviewed drafts of the paper, and approved the final draft.

Ketao Ma conceived and designed the experiments, analyzed the data, prepared figures and/or tables, authored or reviewed drafts of the paper, and approved the final draft.

The following information was supplied relating to ethical approvals (i.e., approving body and any reference numbers):

The Medical Ethics Committee of Shihezi University approved the study (SCXK Xin (2018-001)).

The following information was supplied regarding data availability:

The raw data is available in the Supplemental File.

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
