# Peer review of "Inhibition of Connexin 43 reverses ox-LDL-mediated inhibition of autophagy in VSMC by inhibiting the PI3K/Akt/mTOR signaling pathway"

_PeerJ, doi:10.7717/peerj.12969_

## Round 0.1 · original submission · Major Revisions

As mentioned by the reviewers, the work tackles an important issue, however the conclusions you reached seem to be overstated. Please consider additional experiments that would strengthen your results and support conclusions. When revising the manuscript please pay special attention to the criticism expressed by reviewers 1 and 2. The manuscript must be edited by a professional English editor before resubmitting.

Reviewer 1 ·

Basic reporting

In the manuscript entitled “Connexin 43 participates in ox-LDL-induced autophagy in vascular smooth muscle cells through the PI3K/Akt/mTOR signaling pathway”, by Xuqin Qin and colleagues, it is claimed that Ox-LDL-induced autophagy in VSMC is modulated by Cx43, through the PI3K/Akt/mTOR pathway. It is shown that Gap26, a Cx mimetic peptide corresponding to aa sequence in the first extracellular loop, which is used as gap junction blocker, prevents autophagy inhibition and foam cells formation induced by Ox-LDL. Moreover, it is suggested tht Gap26 effect on autophagy relies on theinhibition of PI3K/Akt/mTOR pathway. This study is biologically interesting and clinically relevant since the unveiling of the mechanisms and signals underlying foam cells formation can open new avenues to design innovative therapeutic approaches against atherosclerosis.
However, the manuscript presents some flaws and weaknesses that limit its novelty and conclusions robustness. Indeed, the conclusions of this study are not supported by the results, which makes it very speculative.
The text seems needs a thorough revisions, in terms of both form and content. Indeed, the sentences are often vague, resorting to terms like “autophagy regulation”, with no description if there is an activation or inhibition of autophagy.

Experimental design

. It is unclear the meaning of “Gap junction channels consist of two corresponding connexins”
. Hemichannels also allow the exchange of molecules between the cytoplasm and extracellular medium.
. Many times, the references used are not appropriate to sustain the sentence. For example, Ref#20 is not the best one to refer to other functions of Cx.
. It should be clearly mentioned that Cx43 can be a substrate of autophagy or a regulatory element of this pathway. This is of utmost importance in the context of this work.
. In M&M section, line 100-105, the two sentences seem similar.
. When using cell cultures, the term “administered” should be replaced by “incubated” or “treated”; also the terms “intervened” or “inoculated” do not sound suitable when referring to cell treatment or seeding.
. The concepts “randomly divided” or “evenly spread” are not appropriate for cell cultures
. The LY294002 group is missing
. Information such as “the six-well plate was removed from the incubator, and the culture medium was discarded” is useless in a scientific publication
. In IF analysis, what the authors mean with “the samples were sealed”?
. the concentration of reagents (PFA, Triton) should be provided in percentage”
. on the other hand, information about TEM analysis is very minimalist; more technical details are needed
. The Oil red O staining protocol description is not helpful; it should be mentioned that was performed according to the manufacturer.
. The results in Fig. 1 and 2 are not new
. To be more accurate, when referring to protein changes, in this context should be used “protein levels” instead of “protein expression.
. Despite the accumulation of mRNA, how can the authors discard the possibility that Cx43 accumulation is not due to autophagy inhibition (Fig.2)? Autophagy flow should be assessed.
. The legend to Fig.2 should include the concentration and time of incubation with OxLDL.
. the IF images are of bad quality, being difficult to discern any change in Cx43 localization. The most important aspect in Cx43 staining should be its localization to the plasma membrane, which is not visible. IF is a technique mainly to assess subcellular distribution and not for protein quantification
. some explanation should be given concerning the characteristics and action of Gap26 and LY294002, as well as the rationale for their use
. The mimetic peptide Gap26 is not a Cx43 blocker; to be more accurate, Gap26 modulates Cx43 channel activity. Are Cx43 levels affected by Gap26?
. the TEM images are of bad quality and not convincing concerning the accumulation of autophagosomes (the use of arrows is not enough); scale bars are missing
. In order to assess the impact/ role of Cx43, the protein should be silenced; when using Gap26 only channel activity is being affected.
. The impact of LY294002 in autophagy has been widely reported
. When referring to protein phosphorylation changes, the term “protein expression” should not be used. The protein expression cannot be affected (protein levels are the same) but the protein undergoes a PTM.
. In line 226, explain the sentence “This finding may be associated with the degradation of internalized Cx43 via activated autophagy”
. In line 226, there is likely a mistake, and “higher” should be replaced by “lower”

Validity of the findings

In general, the results do not support the conclusions

·

Basic reporting

Title of this article should be absolutely revised because the authors demonstrated that blockage of Connexin 43 reversed the autophagy inhibition by ox-LDL. Therefore, the title talking about "Connexin 43 participates in ox-LDL-induced autophagy in VSMCs..." are mis-leading to the audience. I am so confused when reading the whole article.
Accordingly, the title should be revised as "Inhibition of connexin 43 reverses ox-LDL-mediated inhibition of autophagy in VSMCs.....".

Abstract should be revised as well, especially in the conclusion, because "....by mediating the PI3K/AKT/mTOR signaling pathways" are absolutely wrong, you should say "....by inhibition of the PI3K/AKT/mTOR signaling pathways.

English writing should be polished further by native speaker, especially in results part.

Experimental design

The overall experimental design is fine, but the antibodies used in Materials and methods got something wrong. What is anti-Desmin-actin? why need to add "actin" after all name of antibodies?

Validity of the findings

In the results part, the first subheading "Ox-LDL increased p62 protein expression and decreased Beclin-1 protein..." mean nothings. I would suggest to simply the subheading as "Ox-LDL inhibits autophagy induction in VSMCs via downregulation of autophagy-related protein".

In line 209, "To investigate the effect of Cx43 on foam transformation...." should be revised as " To investigate the effect of Cx43 inhibition on form transformation....".

In line 211-213, "These findings is suggested that ..." should be rewritten as "These findings suggested that blocking of Cx43 inhibited the ox-LDL-induced lipid droplet accumulation in VSMCs and suppressed VSMCs foam formation".

In line 217, authors should firstly introduce LY294002 as PI3K inhibitor, otherwise the reader don't know why you use this inhibitor for your assay.

In line 225-226, Cx43 expression in the LY294002 inhibitor group was "lower" instead of "higher" than that in the ox-LDL group.

In line 226-227. The last sentence "This finding may be associated with the degradation of internalized Cx43 via activated autophagy" should be revised because you only demonstrated that Cx43 expression is depended on PI3K signaling by using PI3K inhibitor, how did you show the Cx43 is degraded due to autophagy? Any experimental works to show autophagic degradation of Cx43?

Reviewer 3 ·

Basic reporting

no comment

Experimental design

no comment

Validity of the findings

no comment

Additional comments

Please see the attached PDF

Annotated reviews are not available for download in order to protect the identity of reviewers who chose to remain anonymous.

---

## Round 0.2 · Minor Revisions

Although the manuscript has been improved, Reviewer 1 pointed out that the lack of adequate control experiments makes the conclusions too speculative. Revise please the manuscript and discuss more deeply limitations of the experimental design.In particular, please discuss how knockdown experiments could resolve the conclusions that seem uncertain at this stage of your investigations.

Reviewer 1 ·

Basic reporting

The authors have introduced some changes to the text that helped to improve its quality. However, key experiments that were requested, to support the conclusions were not performed. It is said that these experiments are planned for upcoming studies.

Experimental design

N/A

Validity of the findings

With the lack of additional experiments, namely the silencing of Cx43, some of the conclusions cannot be taken. The use of Gap26 cannot be used as a tool to downregulate Cx43

Additional comments

N/A

Reviewer 3 ·

Basic reporting

No comment

Experimental design

No comment

Validity of the findings

No comment

Additional comments

I am very satisfied with the revisions and recommend this manuscript for publication as is.

---

## Round 0.3 · Minor Revisions

Thank you for adding a discussion on limitations of the experimental design. However, there is one sentence in the revised paragraph that needs to be written more clearly.

Could you please reword the sentence:

"Although some studies have reported that the silencing of Cx43 expression improved the autophagy flux impaired of podocytes (cell line MPC5) induced by high glucose[53], the role of Cx43 in ox-LDL-mediated autophagy of primary VSMC still needs to be verified by more experimental techniques, such as knockdown Cx43 by transfection technology."

Please consider using more descriptive sentences to explain the experiments you are referring to. For example, "In podocytes (MPC5 cell line), silencing of Cx43 expression improved autophagy flow experimentally impaired by exposure of the cells to high glucose levels."

---

## Round 0.4 · accepted · Accept

My last request was handled properly.